# Level of Murine DDX3 RNA Helicase Determines Phenotype Changes of Hepatocytes In Vitro and In Vivo

**DOI:** 10.3390/ijms22136958

**Published:** 2021-06-28

**Authors:** Olga Sergeeva, Tatiana Abakumova, Ilia Kurochkin, Renata Ialchina, Anna Kosyreva, Tatiana Prikazchikova, Varvara Varlamova, Evgeniya Shcherbinina, Timofei Zatsepin

**Affiliations:** 1Skolkovo Institute of Science and Technology, Skolkovo, 121205 Moscow, Russia; T.Abakumova@skoltech.ru (T.A.); I.Kurochkin@skoltech.ru (I.K.); renatayalchina@gmail.com (R.I.); t.prikazchikova@skoltech.ru (T.P.); Varvara.Varlamova@skoltech.ru (V.V.); Evgeniia.Shcherbinina@skoltech.ru (E.S.); t.zatsepin@skoltech.ru (T.Z.); 2Research Institute of Human Morphology, 117418 Moscow, Russia; kosyreva.a@list.ru; 3Department of Chemistry, Lomonosov Moscow State University, 119992 Moscow, Russia

**Keywords:** RNA therapy, transcriptome, RNA helicase, cancer, liver

## Abstract

DDX3 RNA helicase is intensively studied as a therapeutic target due to participation in the replication of some viruses and involvement in cancer progression. Here we used transcriptome analysis to estimate the primary response of hepatocytes to different levels of RNAi-mediated knockdown of DDX3 RNA helicase both in vitro and in vivo. We found that a strong reduction of DDX3 protein (>85%) led to similar changes in vitro and in vivo—deregulation of the cell cycle and Wnt and cadherin pathways. Also, we observed the appearance of dead hepatocytes in the healthy liver and a decrease of cell viability in vitro after prolonged treatment. However, more modest downregulation of the DDX3 protein (60–65%) showed discordant results in vitro and in vivo—similar changes in vitro as in the case of strong knockdown and a different phenotype in vivo. These results demonstrate that the level of DDX3 protein can dramatically influence the cell phenotype in vivo and the decrease of DDX3, for more than 85% leads to cell death in normal tissues, which should be taken into account during the drug development of DDX3 inhibitors.

## 1. Introduction

Recent progress in the consolidation of systems biology, high-throughput methods and bioinformatics has significantly promoted drug development. Early R&D stages are crucial for success in clinical trials as thorough in vitro and in vivo evaluation of new molecules at the preclinical stage provides a solid basis and a margin of safety for the next steps [1,2]. Multiple omics techniques can provide exhaustive data on the changes of RNA, proteins, lipids and metabolites both in vitro and in vivo. Among these methods, transcriptome analysis increases our ability to evaluate the primary response of the cell to any external or internal signal both in vitro and in vivo, thus providing additional valuable information to common cellular assays and long-term animal studies. 

RNA helicases are complex molecular motors that promote structural rearrangements in RNA and RNA-protein complexes and switch their activity [3]. DDX3 helicase is a member of the DEAD-box family, which is involved in many cellular processes, like transport and storage of mRNA [4,5], nuclear export of mRNA [4], mRNA splicing [6], modulation of transcription [7,8], translation [9,10], lipid hemostasis [11], regulation of endoplasmic reticulum stress [12], modulation of epigenetic modifications [13] and innate immune system activation [14]. Also, DDX3 is an essential factor for the replication of some viruses: CMV [15], HIV-1 [16], HCV [17], Japanese Encephalitis virus [18], Dengue virus [19], West Nile virus [20], vaccinia virus [21] and norovirus [22]. Being upregulated in some cancers and due to its contribution toward the replication of several viruses, DDX3 became an attractive target for the development of anticancer and antiviral therapeutics. The first DDX3 inhibitors were studied to target human immunodeficiency virus-1 [23,24]. Then, a number of nucleotide and nucleobase analogs, such as ring expanded nucleosides [25] and other small molecules that target the RNA-binding site [26], were developed as potent anticancer agents. Among them we want to emphasize a tricyclic diimidazodiazepine analog, known as RK-33, that was thoroughly studied due to its broad spectrum of antiviral and anticancer activities [27,28,29]. However, the participation of RNA helicases in almost all aspects of RNA metabolism increases the safety risks of inhibitors and requires thorough evaluation as multiple vital processes can be interfered with [30]. For example, DDX3 can act as an allosteric activator of casein kinase 1 (CK1) in the Wnt/β-catenin pathway [31], TBK1/IKKε and IFN induction [32], NF-κB signal pathway [33], DNA-damage induced apoptosis [34], etc. 

Here, we addressed the question of whether downregulation of DDX3 in the cell can result in significant dysregulation of key cellular processes in vitro (hepatocytes) and in the liver of healthy mice. We compared differential gene expressions after RNAi-mediated downregulation of DDX3 together with histological study. We found that a strong reduction of DDX3 protein (>85%) in comparison to a control leads to similar changes in vitro and in vivo; we observed deregulation of the cell cycle and Wnt and cadherin pathways. As a result, we observed the appearance of dead hepatocytes in the liver and a decrease of cell viability in vitro after prolonged treatment. However, more modest downregulation of DDX3 protein (60–65%) resulted in discordant results between gene expression in vitro and in vivo; in vitro data for both siRNAs were similar, while in vivo, the phenotype was different in comparison to data obtained for the strong reduction of DDX3 protein. These results demonstrate that the amount of DDX3 protein can dramatically influence the phenotype in vivo, which should be taken into account during drug development.

## 2. Materials and Methods

### 2.1. Murine Cell Cultures

Experiments were performed using a Hepa1-6 cell line (ATCC® CRL- 1830, VA, USA). Cells were cultured in DMEM/F12 medium (Gibco™, Waltham, MA, USA), 10% FBS (Gibco™), 1% penicillin and 1% streptomycin (10,000 U/mL, Gibco™) at 37 °C and under 5% CO_2_. Additionally, NIH/3T3 (ATCC® CRL- 1658) and Raw264.7 (ATCC® TIB-71) cells were used for the screening of DDX3 mRNA expression level. NIH/3T3 cells were cultured in DMEM/F12 medium (Gibco™), 10% FBS (Gibco™), 1% penicillin and 1% streptomycin (10,000 U/mL, Gibco™), while Raw264.7 were cultured in DMEM medium (Gibco™), 10% FBS (Gibco™), 1% penicillin and 1% streptomycin (10,000 U/mL, Gibco™) at 37 °C and under 5% CO_2_.

### 2.2. siRNA Design and LNP Formulation

We designed eight chemically modified siRNAs (Appendix A) that targeted murine DDX3 mRNA (NCBI Genbank accession code NM_010028.3). siRNA were selected to avoid off-target binding based on the previously published rules [35,36,37]. In particular, siRNAs were ranked based on the number/positions of the mismatches in the seed region, mismatches in the non-seed region and mismatches in the cleavage site position. The resulting sequences were further filtered to avoid known miRNA motifs and immune stimulatory sequence motifs [38]. Chemical modification of siRNA increases stability against nucleases and further reduces immune response and off-target effects [38,39]. The potency and efficacy of siRNAs were studied by transfection using lipofectamine RNAimax (Thermo Scientific, Waltham, MA, USA) in Hepa1-6 cells followed by reverse transcription-quantitative PCR (RT-qPCR) analysis after 24 h (Appendix A). The control siRNA targets the firefly luciferase gene (control). For the most active siRNAs (#5 and #7), IC50 values were determined. Cells were transfected with 12 concentrations of siRNA; each time, the concentration was decreased three times from 20 nM to 0.04 pM (20 nM, 6 nM, 2 nM, 0.6 nM, etc.) and analyzed for 24 h by RT-qPCR. The IC50 value for siRNA was determined using the Curve Fitting protocol in Graph Pad Prizm version 7.0. Raw data from the RT-qPCR output were plotted against the logarithm of siRNA concentration. The data were normalized from the plateaus of nonlinear regression (Appendix A). 

The most efficient siRNA (#5 and #7) and control siRNA were formulated in lipid nanoparticles (LNPs) as previously described [40,41]. LNPs were dialyzed against PBS at pH7.4 in 20K MWCO cut off dialysis cassettes overnight and filtered through a PES syringe filter (0.2 µm pores). Particle size analysis was carried out using a Zetasizer Nano ZSP (Malvern Panalytical, Worcestershire, UK) according to the manufacturer’s protocol (Appendix A). siRNA entrapment efficiency was determined using the Quant-iT™ RiboGreen® reagent (Thermo Fisher Scientific R11491) as described earlier (Appendix A) [42]. 

We checked the DDX3 mRNA and protein levels after the treatment with control siRNA (siLuc) in vitro and in vivo at the different timepoints (Appendix A) and confirmed that the observed changes were not significant. 

### 2.3. Animal Care and Treatments

All animal care and procedures were carried out according to the relevant National Institute of Health guidelines and were approved by the Bioethics Committee of the Institute of Developmental Biology (Moscow, Russia), where the animal study was performed. Balb/c mice (aged 6–8 weeks) were purchased from Stolbovaya Scientific Center of Biomedical Technologies of the Federal Medical and Biological Agency, Russia. Mice were housed at 22 °C using a 12-h-light to 12-h-dark cycle, fed *ad libitum* with regular rodent chow. Lipid nanoparticles with siRNAs were diluted in sterile saline and injected intravenously *via* the tail vein at the doses and regimes specified in the text (four mice per group). Mice were sacrificed and serum and liver samples were collected for analysis. Serum was collected by cardiac puncture, followed by centrifugation at 1700× *g* for 20 min. Biochemical analysis was performed in Paster Laboratories (Moscow, Russia). Liver samples were snap-frozen, ground and portions of homogenized liver were used for further mRNA and protein analysis.

### 2.4. Histological Analysis 

Liver samples were dehydrated by washing in increasing concentrations of ethanol, followed by xylene and paraffin processing in the automated system Tissue TEK VIP 5 Jr (Sakura, Tokyo, Japan) and filled in blocks in the system Tissue TEK 5 (Sakura). Sections of 8-mm-thick were prepared using rotary microtome Microm GmbH HM 340 (Thermo Scientific, Germering, Germany) and stained with hematoxylin and eosin (H&E) (Abcam, Cambridge, UK) according to the manufacturer's instructions.

### 2.5. RNA Isolation and RT-qPCR

Total RNA was isolated from the cell and liver tissue samples using TRIzol (Thermo Fisher Scientific, Waltham, MA, USA) according to the manufacturer's instructions. Then, ~0.5–1 μg of RNA was further treated with DNase I (Thermo Fisher Scientific) and supplied with RiboLock RNase Inhibitor (40 U/μL) to the final concentration 0.4 U/μL. For RT-qPCR, the treated total RNA was used to synthesize cDNA using a Maxima First Strand cDNA Synthesis Kit (Thermo Fisher Scientific), followed by qPCR using a PowerUp™ SYBR™ Green Master Mix (Thermo Fisher Scientific). PCRs were performed using the primers listed in Appendix A. GAPDH mRNA was used as a control for the analysis of total RNA.

### 2.6. Western Blotting

Cell and tissue protein extracts were prepared from DDX3 KD and control cell and liver samples using RIPA Lysis and Extraction Buffer (Thermo Fisher Scientific) and supplied with 0.05% Triton X-100 (Helicon, Russia), 1 mM dithiothreitol (DTT) (Helicon, Russia), 0.2 mM phenylmethylsulfonylfluoride (PMSF) (Sigma-Aldrich, St. Louis, MO, USA) and 1 x Halt™ Protease Inhibitor Cocktail (Thermo Fisher Scientific). The concentrations of the total protein in the lysates were determined by Bradford protein assay (Thermo Fisher Scientific). Cell extracts (15–50 µg) were denatured by heating in the loading buffer at 95 °C for 10 min and separated by 10% SDS–polyacrylamide gel using PageRuler Pre-stained Protein Ladder as a marker (Thermo Fisher Scientific). Proteins were transferred to nitrocellulose membranes (Bio-Rad) using Mini Trans-Blot® Cell and Criterion™ Blotter (Bio-Rad Laboratories Inc., Hercules, CA, USA) at 80 V for 40 min at RT. The remaining protein-binding sites of the nitrocellulose paper were blocked by immersion in TBS/Tween (10 mM Tris–HCl, 150 mM NaCl, 0.05% Tween 20, pH 7.5) with 5% bovine serum albumin (Sigma-Aldrich) at 4 °C overnight. The blocked filter was incubated with primary antibody anti-DDX3 (A300-476A, Bethyl Lab, 1:3000 dilution), anti-cleaved PARP (ab32064, Abcam, 1:1000), anti-β-Actin (MA1-140, Thermo Fisher Scientific, 1:5000 dilution), anti-cleaved Caspase 3 (9664S, Cell Signaling, 1:1000), p-β-catenin Ser33/37/Thr41 (9561, Cell Signaling, 1:1000) and anti-GAPDH (14C10, Cell Signaling, 1:3000) for 1 h at room temperature. After washing with TBS/Tween, the appropriate secondary antibody was added and incubated for an additional 1 h. Clarity™ Western ECL Blotting Substrates (Bio-Rad Laboratories Inc., CA, USA) were used to develop the blot images. We calculated the Western blot results using ImageJ software according to the standard protocol of the intensity measurement of the bands on the gel (https://imagej.nih.gov/ij/docs/guide/user-guide.pdf (accessed on 1 June 2021)). 

### 2.7. Cell Viability Assay

Hepa1-6 cells were plated into 48-well plates in triplicates, ~25 × 10^3^ cells per well, and transfected with DDX3 siRNA #5 and #7 or control siRNA (final concentration 10 nM). The viability of the cells was measured at 24, 48, 72 and 96 h timepoints after initial transfection. Measurements were done using CellTiter 96® Aqueous One Solution Cell Proliferation Assay (MTS) (Promega, Madison, WI, USA), followed by 3 h incubation at 37 °C. Then, the fluorescent signal was measured using a Varioscan Microplate reader with a 490 nm filter (Thermo Fisher Scientific). 

### 2.8. mRNA-seq Data Processing and Analysis

For transcriptome analysis, we used Hepa1-6 cells after four days of siRNA transfection; three replicates per experiment were used for each siRNA—#5, #7 and control. Total RNA was extracted using a TRIzol reagent (Thermo Fisher Scientific) according to the manufacturer’s instructions. Also, we studied the total RNA sample from the liver tissue after 6 and 13 days of intravenous (IV) LNP-siRNA injections (#5, #7 and control). ~Six micrograms of total RNA (quantified using a NanoDrop™ One C Spectrophotometer (Thermo Fisher Scientific)) was fragmented using conditions optimized to result in an average of 200 nt RNA fragments: incubation for 7 min at 95 °C in RNA fragmentation buffer (100 mM Tris-HCl, 2 mM MgCl2, pH 8.0). The fragmented RNA was purified by precipitation with 96% ethanol/3 M sodium acetate (9:1, v/v), and 1 µg of RNA (measured using a NanoDrop™ One C Spectrophotometer (Thermo Fisher Scientific) was used for an rRNA depletion reaction using a NEBNext rRNA Depletion Kit (NEB E6310L, New England Biolabs, MA, USA) according to the manufacturer’s protocol. Then, the RNA solution was diluted with a 1/10 volume of 3 M sodium acetate; RNA precipitated with ethanol, and 300 ng of RNA (measured using a NanoDrop™ One C Spectrophotometer (Thermo Fisher Scientific) were used for the sequencing library preparation with a NEBNext Ultra II Directional RNA Library Prep Kit for Illumina (NEB 7760, New England Biolabs), according to the manufacturer’s protocol, and the resulting double-stranded cDNA was purified using AMPure XP magnetic beads (A63881, Beckman Coulter, Brea, CA, USA). Efficient concentrations of libraries were determined using RT-qPCR. Library quality (length distribution and the absence of primer-dimers) was assessed by Bioanalyzer 2100 (Agilent Technologies, Netherlands). Libraries were pooled in equal amounts and sequenced using a HiSeq 4000 (Illumina, San Diego, CA, USA) according to the manufacturer’s protocol. Conversion to the fastq format and demultiplexing were performed using bcl2fastq2 software (Illumina, San Diego, CA, USA). In total, 27 samples (DDX3 KD and control KD samples) were sequenced, returning a variable number of paired reads. To map those samples, genome annotations were obtained from Ensembl, release 93. Paired-end reads were mapped using STAR version 2.5.3a [43] with default settings except for the following one: –quantMode GeneCounts. The resulting gene counts were further processed with R package DESeq2 [44], where it was further normalized using the RLE method. In order to take into account unwanted variation in the data, we obtained using the sva package [45] additional variables and introduced them into the design matrix to capture unwanted variations. The DESeq2 package was used for performing differential expression analysis based on the Wald test. We defined genes as differentially expressed if they passed the threshold: FDR < 0.05. For those genes, we performed functional enrichment analysis using cluster Profiler [46]. The method enrichGO was used with BP/MF/CC ontology and BH correction for multiple testing; all genes that were expressed in the mouse were used as the background. Also, we used PANTHER (http://pantherdb.org/ (accessed on 1 June 2021)) pathway analysis. To compare gene expression differences between siRNA #5 and #7 KD sets, we matched gene annotation in the same conditions. Then, we calculated log-fold changes between the gene expression levels in the cells at days 6 and 13 vs a control for both the siRNA #5 and #7 KD sets. To test whether log-fold change values agreed between the two KD experiments, we calculated the Pearson correlation coefficient and performed Fisher’s exact test. Data were submitted to the Gene Expression Omnibus (GEO) (https://www.ncbi.nlm.nih.gov/geo/query/acc.cgi?acc=GSE178618 (accessed on 1 June 2021)) and will be accessible through accession number GSE178618.

### 2.9. Statistical Analysis of the Experimental Data

All diagrams are based on at least three independent experiments. Statistical data processing was performed using the GraphPad Prism software (version 8.3) (Graphpad Holdings, LLC, CA, USA) with a two-sample t-test, as well as a two-way ANOVA analysis of variance or repeated-measures ANOVA and Sidak t-test. The data were considered statistically significant at *p* < 0.05.

## 3. Results

### 3.1. Knockdown of DDX3 RNA Helicase in vitro

We compared DDX3 mRNA levels in murine fibroblast (NIH/3T3), macrophage (RAW 264.7) and hepatoma (Hepa1-6) cell lines and confirmed the high level of mRNA in hepatoma cells (Appendix A). In order to achieve silencing of DDX3 mRNA, we designed and synthesized eight siRNA and a control siRNA targeting the firefly luciferase gene (marked as Luc control) and screened them in Hepa1-6 cells (Appendix A). For the two most efficient siRNAs (#5 and #7), we determined IC_50_ as the use of siRNA at low concentrations additionally decreases the possibility of off-target effects in vitro and in vivo. For siRNA #5, the IC_50_ was 8±1.5 pM; for #7, it was 3.2±0.4 pM (Appendix A). These potent siRNA were used in further studies. Also, we determined the level of DDX3 protein after 1, 2, 3 and 4 days of mRNA downregulation by siRNA #5 in Hepa1-6 cells and found that at day 4 the efficacy of DDX3 protein downregulation was ~80% (Appendix A). The efficacy of DDX3 mRNA and protein downregulation by siRNA #5 and #7 in vitro was comparable with published results [47,48,49].

### 3.2. Knockdown of DDX3 RNA Helicase in vivo

First, we estimated DDX3 mRNA expression levels in different murine organs (Appendix A). DDX3 RNA helicase was expressed in the heart, liver, kidneys, testicles, muscles and brain at a similar level, while in the spleen and lungs the mRNA expression was higher. Nevertheless, the DDX3 level in the liver was high enough to perform the study. Selected DDX3 siRNA were formulated into C12-200 lipid nanoparticles, previously validated in mice and nonhuman primates [50]. Due to their size (80–90 nm) and almost neutral charge, C12-200 siRNA-LNPs are preferably internalized by hepatocytes. Biodistribution of C12-200 siRNA LNP after intravenous injection in mice has been thoroughly assessed in previous studies, confirming liver-specific RNAi-mediated silencing [51]. 

We performed dose-response and mRNA recovery experiments for LNP-siRNA #5 and #7 to find an optimal regime for our study (Appendix A). A single administration of 1 mg/kg LNP-siRNA #5 and #7 resulted in the profound knockdown of DDX3 mRNA in the liver with different efficacies. By 72 h post-injection, siRNA #5 was more active and resulted in 75–80% of mRNA downregulation, whereas siRNA #7 was less active, resulting in 60–65% of mRNA downregulation in comparison to the DDX3 mRNA level in the liver without siRNA treatment (Appendix A). Maximal mRNA and protein downregulation in the liver occurred two days after injection, followed by a slow recovery (Appendix A). Hence, DDX3 silencing >50% lasted for at least eight days. 

After evaluation of DDX3 knockdown with the single dose of LNP-siRNA, we studied the toxicity of LNP-siRNA. Balb/c females received biweekly injections of LNPs loaded with siRNA (1 mg/kg) for three weeks. Analysis of the animal blood demonstrated that levels of ALT, AST, ALP and other key factors were not changed. Also, no significant weight changes or differences in the behavior of the mice were observed between the groups (Appendix A), indicating that long-term administration of LNP-siDDX3 was well tolerated. 

Then, we studied the level of DDX3 mRNA at days 6, 8 and 13 for biweekly injections of 1 mg/kg LNP-siRNA (Figure 1A). We found that siRNA #5 had higher activity in comparison to siRNA #7, which correlated with the results in vitro (Figure 1B). Initial downregulation of DDX3 protein at day six was comparable: ~75% for siRNA #5 and ~60% for siRNA #7. However, the further trend for DDX3 protein level was different for siRNA #5 and #7 (Figure 1C). At day eight, the efficacy of protein downregulation was still similar for both siRNAs (75–80%, while more pronounced for siRNA #5), but at day 13, the level of downregulation of DDX3 protein changed. For siRNA #5, the DDX3 protein level dropped to <15%, while for siRNA #7, it recovered to ~35%. As a result, we selected days 6 and 13 as timepoints for further gene expression analysis after the knockdown of DDX3 RNA helicase in vivo.

### 3.3. Efficient Knockdown of DDX3 in the Murine Liver and Hepa1-6 Cells Induces Apoptosis of Hepatocytes

To explore the effects of DDX3 downregulation during long-term administration (6 and 13 days) of LNP-siRNA in the murine liver, we performed a morphological study using hematoxylin-eosin (H&E) staining of liver samples. According to our results, no pathological changes were observed in the Luc control group of Balb/c mice. Administration of more active siRNA #5 resulted in minor focal accumulation of lymphocytes, neutrophils and histiocytes in hepatic lobules at day six. Also, all females had an increased number of non-epithelial cell elements along the beams. At day 13, the number of small foci of neutrophil infiltration increased, and also, we observed more binuclear hepatocytes. Two mice had individual dying hepatocytes and few minor foci of neutrophil infiltration in the stroma, along with focal necrosis. Administration of less active siRNA #7 resulted in more soft effects. We still observed an increased number of binucleated hepatocytes, along with some accumulation of lymphocytes, neutrophils and histiocytes at days 6 and 13, but these changes were less significant than in the case of siRNA #5 and no dying cells were observed (Figure 2A).

To reveal the driving force of hepatocyte death in the liver of DDX3 KD mice, we analyzed the level of cleaved caspase-3, a key player in apoptosis. We found the activation of caspase-3 after the administration of LNP-siRNA #5 for 13 days, which correlates with the hepatocyte death detected by morphological study (Figure 2B). At the same time, the level of the cleaved caspase-3 was not increased after administration of less effective LNP-siRNA #7. 

To study the influence of DDX3 protein level on the survival of hepatocytes, we performed MTS assay using Hepa1-6 with DDX3 KD and control cells (Figure 3A). We observed a slight decrease in cell survival after DDX3 mRNA KD during the first three days. Hence, the behavior of the survival curve was similar for siRNA #5 and #7. However, at day four after administration, we observed the decrease in cell survival for siRNA #5, while for siRNA #7, no effects on cell survival were observed. Then, we measured the level of a hallmark of apoptosis, a cleaved form of PARP protein. We found that cleaved PARP was increased in the case of siRNA #5 KD, while in the case of siRNA #7, KD changes were more modest (Figure 3B). Activation of apoptosis in hepatocytes after prolonged siRNA #5 administration can explain the decrease in cell survival. At the same time, a slower increase in cleaved PARP after siRNA #7 administration did not activate cell apoptosis during the first three days of knockdown.

### 3.4. Analysis of Gene Expression after Knockdown of DDX3 in vitro and in vivo

RNA sequencing was used to profile the mRNA expression after RNAi-mediated downregulation of DDX3 RNA helicase in vitro and in vivo. We analyzed changes in mRNA expression at day four after DDX3 knockdown in vitro by more active siRNA #5 and less active siRNA #7. The same approach was used in vivo: we analyzed the transcriptome of the murine liver at days 6 and 13 after DDX3 KD by LNP-siRNA #5 and #7. Comparison of scatterplots of the changes of mRNAs expression levels revealed a huge difference between the phenotypes of DDX3 KD cells obtained with siRNA #5 and #7 in vitro and in vivo because many changes did not overlap (Figure 4A).

We analyzed changes in mRNA expression after siRNA #5 administration in vitro and found 6424 changed mRNA in comparison to control siRNA (cutoff 2-fold, *p* < 0.05), which included 3418 upregulated and 3006 downregulated ones (the list is shown in Appendix A). In the case of siRNA #7 DDX3 KD in vitro, we found only 3104 changed mRNA in comparison to control siRNA, which included 1781 upregulated and 1323 downregulated ones. After administration of LNP-siRNA #5 in vivo at day six, we found 2116 changed mRNA in comparison to control siRNA, among which 929 ones were upregulated and 1187 downregulated. At day 13. some increase was observed; the expressions of 2974 mRNA were changed in comparison with control siRNA, including 1566 upregulated and 1408 downregulated ones. After administration of LNP-siRNA #7 in vivo at day six, only 460 mRNA were changed in comparison with control siRNA; among them, 260 were upregulated and 200 were downregulated. At day 13, some increase was observed: 782 mRNA were changed in comparison with control siRNA; 474 were upregulated and 308 downregulated. So, the use of more active siRNA #5 to knockdown DDX3 in comparison to less effective siRNA #7 led to changes in the expression of much more mRNA both in vitro and in vivo (Figure 4B). Based on the volcano plot data and exact mRNA with change expression, the phenotypes of murine livers with DDX3 KD inhibition significantly differed from hepatocytes in vitro (Appendix A). 

### 3.5. GO Pathway Enrichment Analysis

We performed PANTHER pathway enrichment analysis (http://pantherdb.org/ (accessed on 16 February 2021)) of changed mRNA after DDX3 KD in vitro and in vivo. PANTHER is a database that combines gene functions, ontology, pathways and statistical analysis tools and allows enriched pathways for selected genes to be analyzed.

We found that knockdown of DDX3 mRNA in Hepa1-6 cells in vitro by siRNA #5 caused similar changes as siRNA #7 in the pathways determined by PANTHER. Among them were the Wnt, CCKR, EGF receptor, interleukin and cadherin signaling pathways (Figure 5). 

The phenotype after DDX3 knockdown with siRNA #7 in vivo differs a lot from the phenotype in vitro. At day six of DDX3 KD, less active siRNA #7 caused reliable changes only in the biosynthesis of cholesterol and glutamate receptor group pathways. At day 13, we observed changes in Wnt and inflammation signaling pathways mediated by chemokine and cytokine. Administration of more active siRNA #5 led to more significant changes in pathways, including Wnt, integrin, p53 and cell cycle at day six. To evaluate the obtained PANTHER pathways and confirm the transcriptome data, we performed RT-PCR analysis of the genes involved in cell cycle regulation, the Wnt signaling pathway and glucose and lipid metabolism after DDX3 KD in vitro and in vivo (Figure 6A). The observed changes correlate with the data from transcriptome analysis and additionally confirm the results of the PANTHER analysis. For additional investigation of the Wnt signaling pathway in vivo, we checked the phosphorylation level of β-catenin and found a decrease at day 13 in the murine liver after DDX3 KD (Figure 6B). Changes of β-catenin phosphorylation indicate the activation of the Wnt pathway.

Prolonged siRNA #5 action led to the activation of many cellular signaling pathways at day 13: integrin, Wnt, cadherin, Hedgehog, PDGF, endothelin and glycolysis. Based on these results, the phenotype in vivo after prolonged administration of more active siRNA #5 (13 days) caused the same transcriptome changes of the pathways as both siRNAs in vitro. siRNA #7 was less active in vivo and caused quite different phenotype changes (Figure 5).

Then we used gene ontology KEGG analysis, a common tool to study the functional relationship between gene products, and predicted three aspects: biological processes (BP), cellular components (CC) and molecular functions (MF) (Appendix A). Analysis of in vitro and in vivo data shows high similarity to results obtained by the PANTHER tool. The phenotypes of siRNA #5 and siRNA #7 in the cells were similar and caused the same changes in BP, CC and MF. Hence, DDX3 knockdown in vivo differed a lot from siRNA #7, which caused less significant changes in comparison to siRNA #5. Prolonged administration of siRNA #5 led to phenotype changes in vivo close to those in vitro.

## 4. Discussion

DDX3 RNA helicase was intensively studied as a therapeutic target due to its contributions to the replication of viruses and cancer progression [52,53]. At the same time, DDX3 participates in almost all aspects of cell life, such as transcription [54,55,56], pre-mRNA splicing [57], mRNA translation [58,59,60], cell cycle regulation [61] and apoptosis [62]. As a result, there are reports that propose either oncogenic or tumor suppressor functions for DDX3 depending on the cancer, cell type or xenograft model used in the studies [57]. The small-molecule inhibitor of DDX3, RK-33, decreases the proliferation of multiple lung cancer cell lines in a dose-dependent manner and acts as a radiosensitizer in mouse models of lung cancer [29]. Bol et al. [28] demonstrated that changes in the human cancer lung cell line after RK-33 and shRNA-mediated DDX3 protein inhibition were similar: G1 cell cycle arrest, deregulation of Wnt pathway and induction of apoptosis. Also, after RK-33 administration in the xenograft murine model of prostate cancer, the level of caspase-3 was increased and H&E staining revealed increased cell death [29]. However, this result was obtained 24 h after RK33 administration and no data on the long-term treatment or the influence of DDX3 inhibition on healthy tissues was presented. Here, we addressed the safety issues of targeting DDX3 RNA helicase by a comparison of the gene expression in the cancer hepatocyte Hepa1-6 cell line and the liver of healthy mice after short and long-term DDX3 knockdown. We selected an RNAi-based approach because it can be applied both in vitro and in vivo and we can perform long-term downregulation of the target without significant toxicity and off-target effects. A number of siRNA drugs are already approved by the FDA and EMA, including patisiran [63], with siRNA embedded in lipid nanoparticles (LNP) for targeted delivery to the liver. We formulated active siRNAs in similar lipid nanoparticles for targeted delivery in the murine liver [42].

First, we selected potent and efficient siRNA (#5 and #7). They demonstrated similar activity in the cell line (~80% knockdown of DDX3 mRNA and protein; see Appendix A), while in vivo we observed a significant difference for siRNA: #5 was more active, while #7 less active (Figure 1C). As the siRNA demonstrated close efficacy at day six and the most pronounced difference at day 13, we chose these timepoints for further morphological analysis and deep RNA sequencing of liver samples in comparison to controls and DDX3 knockdown Hepa1-6 cells in vitro. At day 13 of DDX3 knockdown, H&E staining of the liver revealed infiltrated immune cells after administration of both siRNA #5 and #7 and the appearance of individual dying hepatocytes in the case of siRNA #5 (Figure 2A). The appearance of individual apoptotic hepatocytes after long-term administration of the highly active siRNA #5 confirms the damage to healthy cells. We additionally demonstrated the activation of caspase-3 cleavage after long-term administration of the most effective siRNA#5 in vivo (Figure 2B). All these data correlate with the results demonstrated for RK-33 treatment [29]. 

Also, we checked the cell proliferation and protein level of cleaved PARP and found that more active siRNA #5 caused a reduction in the proliferation and induction of apoptosis in vitro (Figure 3). This cellular phenotype after RNAi-mediated knockdown corresponds to the data obtained for the hepatocyte-specific DDX3 knockout mice [64]. We want to emphasize that after administration of siRNA #7 with lower knockdown, similar effects were not observed. To compare changes in gene expression under these conditions, we performed transcriptome analysis of DDX3 knockdown Hepa1-6 cell in vitro and liver samples at days 6 and 13 of DDX3 KD to uncover dynamic response to varied levels of DDX3 protein.

As we expected, more active siRNA #5 caused more significant changes in gene expression than siRNA #7 both in vitro and in vivo (Figure 4). Gene function analysis with the PANTHER classification system of DDX3 knockdown in vitro demonstrated deregulation of many cell signaling pathways: EGF, cadherin, PDGF, cholesterol biosynthesis and Wnt (Figure 5). It was shown that DDX3 regulates cell migration and invasion through an E-cadherin mediated pathway [65,66], is involved in the Wnt/β-catenin signaling pathway and can affect the Wnt cascade [67]. In the case of cancerous cells, these changes are beneficial for treatment and correspond to the phenotype after DDX3 inhibition in different cell types [31,67,68,69]. However, DDX3 knockdown in vivo resulted in different changes in gene expression for siRNA #5 and #7. At day six (siRNA #5), we observed differential expression of genes connected with the p53 pathway, integrin, cell cycle and Wnt pathways (top 5) (Figure 5 and Figure 6) that correlates with our in vitro data and published data [23,59,64,66,67,70]. In the case of siRNA #7, we can propose that 35–40% of the residual DDX3 protein is enough for normal cell functioning as only two pathways were changed in the murine liver. Long-term administration for 13 days of siRNA #7 caused changes in the inflammation-mediated and Wnt pathways. Long-term administration of siRNA #5 led to changes of gene expression in most cellular signaling pathways, which could be the result of multiple secondary effects (Figure 5 and Figure 6). 

Comparison of RNA levels (transcriptome data) between in vitro and in vivo studies can reveal discordant gene expression even under close conditions [71,72]. This difference may be determined by varied inhibition or downregulation of the target in these conditions. Organ or tissue homeostasis in vivo is more complex and includes immune responses, metabolic activation, hormone signaling and other regulatory events in comparison to common cell culture [73]. However, in the case of strong DDX3 KD, we got similar changes in PANTHER pathways in vitro and in vivo, and the phenotype after siRNA #5-mediated DDX3 knockdown is similar to one observed after RK-33-mediated DDX3 inhibition [28,29]. Most of RK-33 studies were short-term and were done in cancer cells or xenograft mice models. Our results in the normal murine liver demonstrated that strong DDX3 knockdown (more than 80%) caused changes in the main cellular pathways and finally led to the appearance of dying hepatocytes. Long-term treatment with less active siRNA #7 did not cause such dramatic changes of the hepatocyte phenotype and cell death. 

There are many examples when RNAi results in non-specific effects often referred to as off-target gene silencing [74,75]. Of course, a more reliable approach to confirm the knockdown data is one using multiple siRNAs. At the same time, this strategy can lead to the increase of non-specific effects, especially in the case of targets that participate in many biological processes in the cell. Previously, it was demonstrated that appropriate chemical modification and high efficacy of siRNA potently reduce potential off-targets [76,77]. In all investigations based on transient knockdown of mRNA, there is a probability of non-specific effects, but if the obtained data correlate with each other and with the published data, we can conclude that most of them are reliable. The results for DDX3 KD are consistent with each other: the phenotype of the KD cell line was the same, and the changes of transcriptomes in vitro and in vivo were similar in biological processes for siRNA#5. One more proof of reliability comes from similar transcriptome changes for siRNA#7 KD in vivo. However, these changes were observed later (at day 13) and to a smaller extent. So, based on the data convergence coming from on-target-related changes in biological pathways using two siRNAs, we conclude that DDX3 protein contribution to various processes may depend on the level of the protein in the cell. 

Here, we demonstrated protein level-dependent effects of DDX3 RNA helicase knockdown on the murine liver transcriptome. Strong downregulation of DDX3 led to the phenotype close to the published one, obtained with the small molecule inhibitor RK-33. However, these changes are beneficial only for cancer cells, while we found deregulation of many cellular processes and the appearance of the single dying hepatocytes in healthy livers. These follow-up events may be crucial for the successful development of therapeutics targeting the DDX3 RNA helicase.

## Figures and Tables

**Figure 1 ijms-22-06958-f001:**
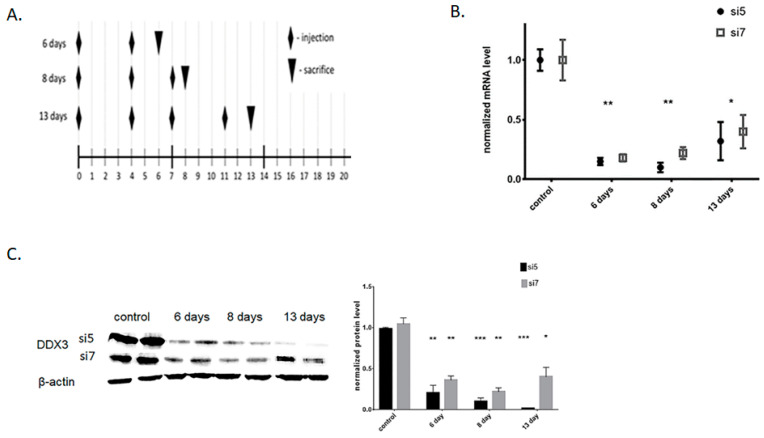
Knockdown of DDX3 in murine liver. (**A**) Schedule of LNP-siRNA injections in vivo. (**B**) RT-qPCR quantification of DDX3 mRNA recovery after single LNP-siRNA #5 and #7 IV injection—mRNA levels at days 6, 8 and 13. (**C**) Western-Blot analysis of DDX3 protein at days 6, 8 and 13 after multiple LNP-siRNA #5 and LNP-siRNA #7 injections according to schedule 1A. * *p* < 0.05, ** *p* < 0.01, *** *p* < 0.001.

**Figure 2 ijms-22-06958-f002:**
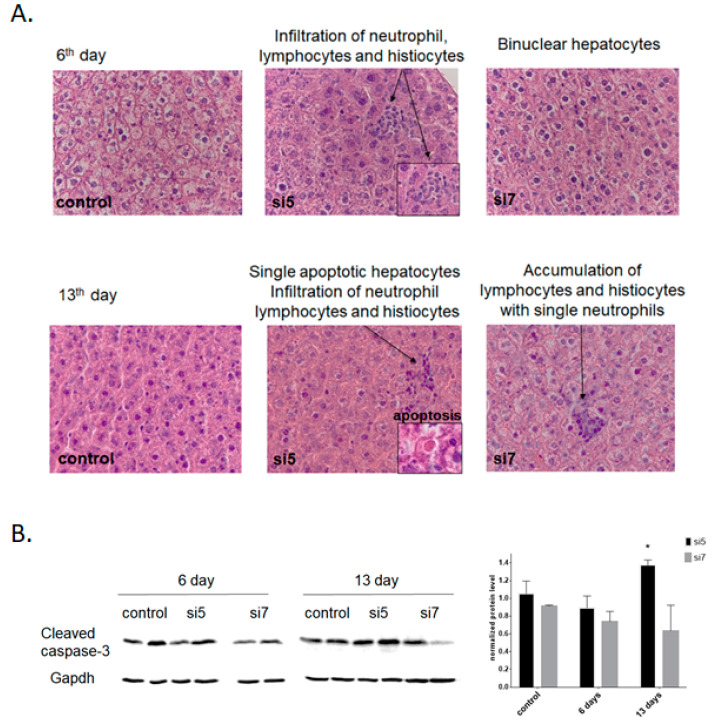
Characterization of murine liver phenotype after DDX3 knockdown. (**A**) Morphological analysis (H&E staining) of liver samples after DDX3 knockdown with LNP-siRNA #5 and #7 at days 6 and 13. (**B**) Quantification of cleaved caspase-3 in DDX3 KD hepatocytes at days 6 and 13 after initial siRNA transfection (normalized to control cells). * *p* < 0.05.

**Figure 3 ijms-22-06958-f003:**
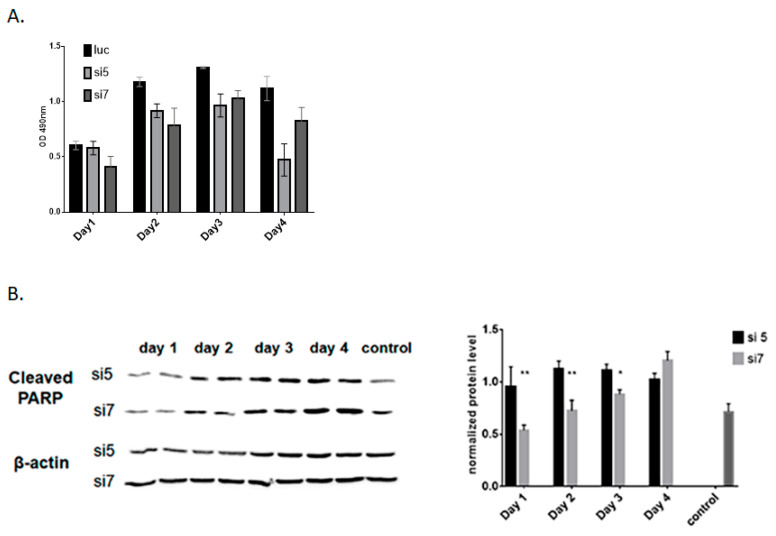
Analysis of Hepa1-6 cells after DDX3 knockdown in vitro. (**A**) Viability assay of DDX3 depleted cells using siRNA #5 and #7 at days 1, 2, 3 and 4 after initial siRNA transfection (normalized to control cells and viability at day 1). (**B**) Quantification of cleaved PARP in DDX3 KD hepatocytes at day 1, 2, 3 and 4 after initial siRNA transfection (normalized to control cells). * *p* < 0.05, ** *p* < 0.01.

**Figure 4 ijms-22-06958-f004:**
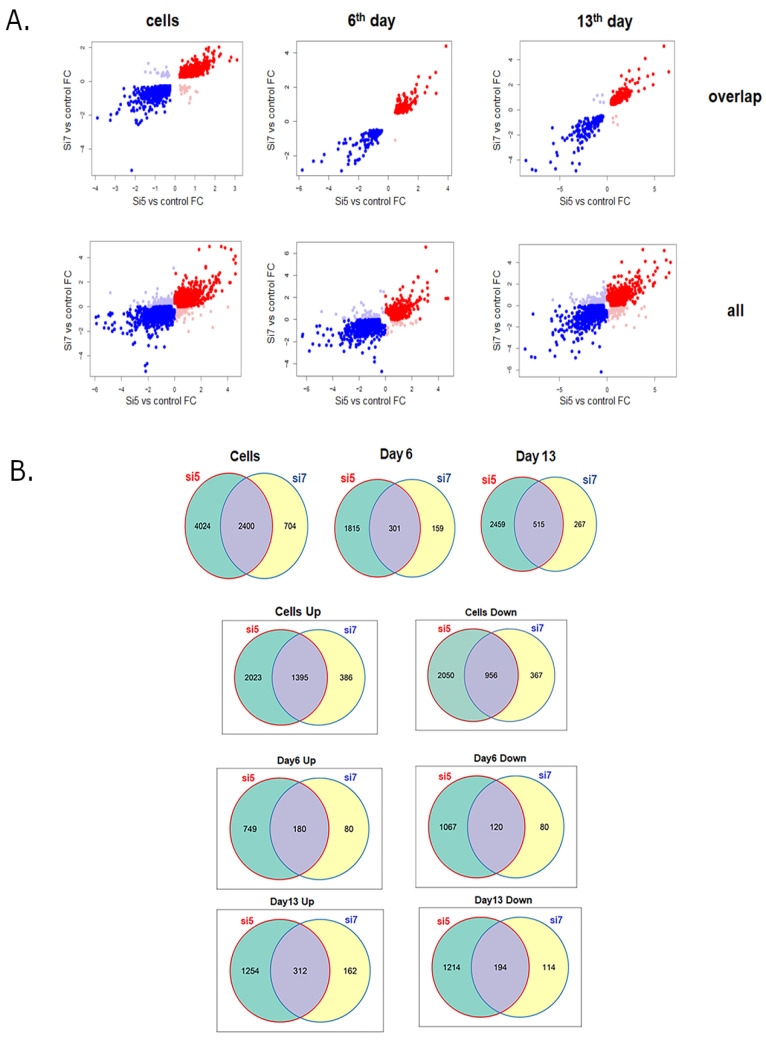
Comparison of differential gene expressions between DDX3 KD using siRNA #5 and #7 in hepatocytes in vitro, and in the liver at days 6 and 13 after LNP-siRNA delivery. (**A**) Scatterplots of the expression differences after siRNA#5 and siRNA#7 in comparison to controls (shown as log fold change). Dots represent individual genes that differ in both conditions. Colors indicate plot quadrants. (**B**) Venn diagram analysis of gene expression changes after DDX3 KD with either siRNA #5 or #7.

**Figure 5 ijms-22-06958-f005:**
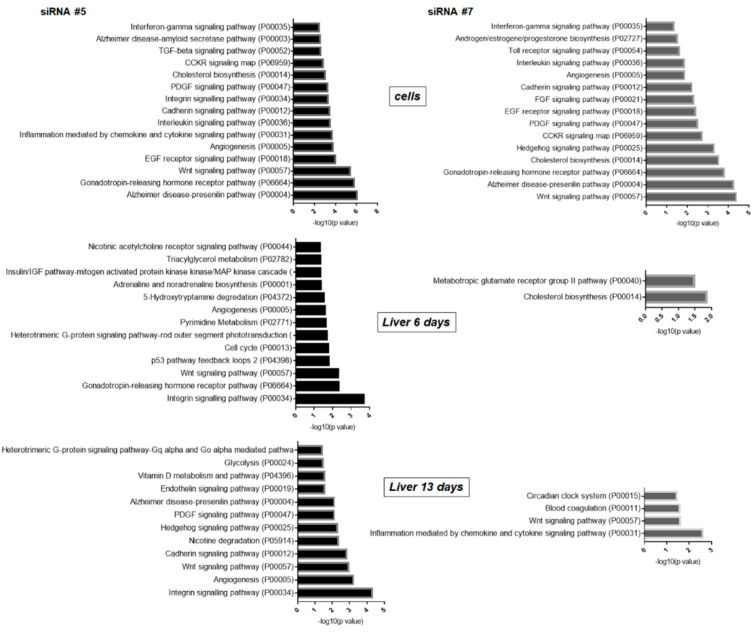
Top representative pathways obtained by PANTHER analysis of transcriptomes: DDX3 KD with siRNA #5 and #7 in hepatocytes in vitro, in the liver at days 6 and 13 after LNP-siRNA delivery.

**Figure 6 ijms-22-06958-f006:**
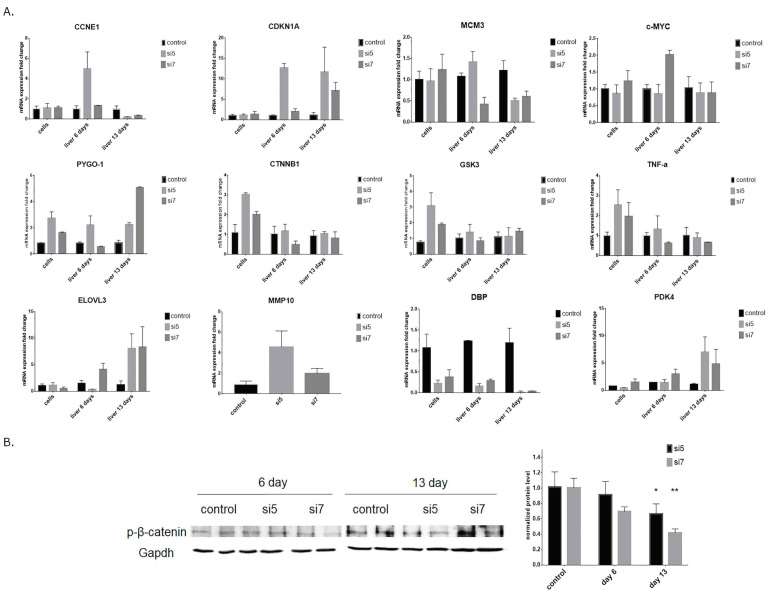
(**A**) Confirmation of the in vitro and in vivo transcriptome data by RT-qPCR after DDX3 KD with siRNA #5 and #7 by quantification of mRNA expression fold change in comparison with control siRNA. (**B**) Quantification of the b-catenin phosphorylation level in DDX3 KD in murine liver. Mmp10 mRNA was measured only in the cells. * *p* < 0.05, ** *p* < 0.01.

## Data Availability

Publicly available datasets were analyzed in this study. This data can be found here: https://www.ncbi.nlm.nih.gov/geo/query/acc.cgi?acc=GSE178618 (accessed on 1 June 2021) (GEO accession GSE178618).

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
