# Peer review of "Level of Murine DDX3 RNA Helicase Determines Phenotype Changes of Hepatocytes In Vitro and In Vivo"

_ijms, 2021, doi:10.3390/ijms22136958_

Round 1

Reviewer 1 Report

Specific comments:

  • The current title, “Eccessive suppression of murine DDX3 RNA helicase is negative for hepatocytes in vitro and in vivo,” – did not precisely reflect the data.
  • Lines 11-12: “we selected siRNAs with different efficacy and demonstrated in vivo protein level-dependent effects in DDX3-depleted.” It is not logical.
  • Lines 13-14: “We used transcriptome analysis to estimate the primary response of the hepatocytes to the knockdown of DDX3 RNA helicase both in vitro and in vivo.” It is not a complete sentence.
  • Lines 15-16: They did not validate the data “deregulation of cell cycle, Wnt and cadherin pathways.”
  • Line 19: “strong knockdown and a weak phenotype in vivo” – neither is well defined.
  • Line 20: “level of active DDX3 protein” – what is “active” or “excessive decrease of DDX3” vs. “strong” or “reduction?” They did not define these terms, including what level of protein or how active is required for normal physiology. Specifically, you may have 100% protein, but only 1% activity is sufficient. These protein and activity are two different concepts – the authors mixed them up and freely exchanged them in the text -
  • Lines 71-75: How long did they use the cell lines before acquiring them? What was the QC to confirm the identity? What did they use hepatocytes and the control cells to screen the DDX3 mRNA expression level?
  • Lines 95-100: What was the QC for LNP?
  • Lines 102-114: How many mice did they use per group?
  • Line 214: “cancerous hepatocytes” vs. “hepatoma (Hepa 1-6) cell line” – what was the feature of both lines?
  • Lines 214-216: “In order to achieve silencing of DDX3 mRNA, we designed and synthesized eight siRNA and a control siRNA targeting the firefly luciferase gene (marked as Luc control) and screened them in Hepa1-6 cells (Suppl. Fig. 1B).” why did they use Luc?
  • Lines 108-109: “Lipid nanoparticles with siRNAs were diluted in sterile saline and injected intravenously via tail vein at doses and regimes specified in the text.” What were mechanisms that impact the heart, liver, kidney, testicles, muscles, and brain (Lines 205-209) with such non-specifically systemic delivery? What were the data for those organs?
  • Fig 1. How did they determine the dosages? What was the organ? Lines 262-263: “multiple 262

LNP-siRNA #5 and #7 injections” – multiple? Or #5 and #7 injections? Out of how many mice did they observe?

  • Fig 2. Any alternative confirmation?
  • Fig 3. Any side-by-side comparison with in vivo? What did they mean using error bars? P-values?
  • The text used either siRNA #5 or #7; however, they used si1 or si2 in the supplemental information.
  • Line 329 “differed from hepatocytes in vitro (Suppl. Fig. 4 A-C).” Where are those Suppl. Figs?
  • Lines 358-359: “Prolonged administration of siRNA #5 lead to phenotype changes in vivo close to those in vitro.” What did they mean “to phenotype changes in vivo” in what ways? Any data?
  • Lines 404-405: “more active siRNA #5 caused more significant changes in gene expression than siRNA #7 both in vitro and in vivo (Fig. 4).” - They said multiple, but only two listed – what happened to others?
  • Lines 436-437: “Here we demonstrated protein level-dependent effects on DDX3 RNA helicase knockdown in the murine liver.” What was about the RNA seq data set?
  • Precaution is needed for spelling and English expression patterning, e.g., title “Eccessive suppression of murine DDX3 RNA helicase is negative for hepatocytes in vitro and in vivo” should be Excessive.

Author Response

Thank you for your evaluation of the manuscript and providing valuable comments and suggestions. We addressed all recommendations as explained below. All our corrections in the text indicated in yellow.

Specific comments:

  • The current title, “Eccessive suppression of murine DDX3 RNA helicase is negative for hepatocytes in vitro and in vivo,” – did not precisely reflect the data.

We have changed the title of the paper to “Level of murine DDX3 RNA helicase determines phenotype changes of hepatocytes in vitro and in vivo”, and expect that current title is consistent with the presented data.

  • Lines 11-12: “we selected siRNAs with different efficacy and demonstrated in vivo protein level-dependent effects in DDX3-depleted.” It is not logical.

We deleted this phrase and modified the next one to: “We used transcriptome analysis to estimate a primary response of the hepatocytes to different efficacy of RNAi-mediated knockdown of DDX3 RNA helicase both in vitro and in vivo”. Initial phrase was shorted to fit the word limit of the abstract that led to misunderstanding

  • Lines 13-14: “We used transcriptome analysis to estimate the primary response of the hepatocytes to the knockdown of DDX3 RNA helicase both in vitro and in vivo.” It is not a complete sentence.

We modified the phrase.

  • Lines 15-16: They did not validate the data “deregulation of cell cycle, Wnt and cadherin pathways.”

We performed these experiments and provided data in the text (Fig. 6 and Fig. 2B).

  • Line 19: “strong knockdown and a weak phenotype in vivo” – neither is well defined.

We rephrased the sentence.

  • Line 20: “level of active DDX3 protein” – what is “active” or “excessive decrease of DDX3” vs. “strong” or “reduction?” They did not define these terms, including what level of protein or how active is required for normal physiology. Specifically, you may have 100% protein, but only 1% activity is sufficient. These protein and activity are two different concepts – the authors mixed them up and freely exchanged them in the text –

We changed the sentence to “These results demonstrate that the level of DDX3 protein can dramatically influence on the cell phenotype in vivo and efficacy decrease of DDX3 more than 85% leads to the cell death in normal tissues, which should be taken into account during drug development of DDX3 inhibitors”.

  • Lines 71-75: How long did they use the cell lines before acquiring them? What was the QC to confirm the identity? What did they use hepatocytes and the control cells to screen the DDX3 mRNA expression level?

We purchased the cells in ATCC in 2018, prepared cell stocks and used each of them for less than 10 passages. Also, before each experiment we checked the cell morphology by microscopy. We use efficient LNP-based technology for siRNA delivery to hepatocytes in vivo, so in this study we first checked DDX3 mRNA expression level in hepatocytes and then moved to in vivo studies.

  • Lines 95-100: What was the QC for LNP?

We determined the size and PDI for LNP and quantified siRNA loading to LNP. We added this information in the Supplementary material – Supplementary table 3.

  • Lines 102-114: How many mice did they use per group?

We used 4 mice in the group. We added the information in the “Materials and methods” section.

  • Line 214: “cancerous hepatocytes” vs. “hepatoma (Hepa 1-6) cell line” – what was the feature of both lines?

Thank you for pointing out the inaccuracy of the wording. We made changes.

  • Lines 214-216: “In order to achieve silencing of DDX3 mRNA, we designed and synthesized eight siRNA and a control siRNA targeting the firefly luciferase gene (marked as Luc control) and screened them in Hepa1-6 cells (Suppl. Fig. 1B).” why did they use Luc?

Renilla luciferase mRNA is not expressed in human cells, so siRNA targeting this mRNA is a generally accepted and validated control [C. Curtis et al, 2014; A. Raven et al, 2017; D. Leboeuf et al, 2020 and others]. At the same time, using LNP with formulated siRNA is a better control then just PBS or untreated mice.

  • Lines 108-109: “Lipid nanoparticles with siRNAs were diluted in sterile saline and injected intravenously via tail vein at doses and regimes specified in the text.” What were mechanisms that impact the heart, liver, kidney, testicles, muscles, and brain (Lines 205-209) with such non-specifically systemic delivery? What were the data for those organs?

Previously it was shown that these lipid nanoparticles (lipid/lipidoid composition, size, zeta potential) are delivered preferably to the liver without affecting other organs, links # 40 and 51. Due to peculiarities of endosomal escape in different tissues and cell types these siRNA-LNP mainly lead to the target mRNA downregulation in the liver [K. Love et al, 2010; K. Whitehead et al, 2014].

  • Fig 1. How did they determine the dosages? What was the organ? Lines 262-263: “multiple 262

LNP-siRNA #5 and #7 injections” – multiple? Or #5 and #7 injections? Out of how many mice did they observe?

We did the dose-optimization experiment: injected 0,5 mg/kg, 1 mg/kg and 2 mg/kg and measured the efficacy of DDX3 mRNA inhibition in murine liver (Suppl. Fig. 2A). Results demonstrated that 1 mg/kg was the minimal optimal dose. Also we explained before that we measured the efficacy of KD in the liver because “Biodistribution of C12-200 siRNA LNP after intravenous injection in mice has been thoroughly assessed in previous studies, confirming liver-specific RNAi-mediated silencing [51]”. Fig. 1C demonstrated the results of DDX3 protein analysis by WB after the multiple injections by LNP-siRNA#5 and LNP-siRNA#7 according to the scheme showed on the Fig. 1A.

  • Fig 2. Any alternative confirmation?

We added additional experimental data (Fig. 2B).

  • Fig 3. Any side-by-side comparison with in vivo? What did they mean using error bars? P-values?

Fig. 3 described the comparison for the Hepa1-6 cell line. For the in vivo work we also demonstrated that only after administration of LNP with the most efficient siRNA#5 there was the activation of the apoptosis and appearance of single died hepatocytes only after prolonged DDX3 inhibition for 13 days (Fig. 2). Error bars demonstrated the standard deviation.

  • The text used either siRNA #5 or #7; however, they used si1 or si2 in the supplemental information.

We used other names only in the Suppl. Fig. 5 because it was done by standard software of our bioinformatics collaborator and the program renamed siRNA#5 as lnc1 and siRNA#7 as lnc2. We marked this point in the figure description.

  • Line 329 “differed from hepatocytes in vitro (Suppl. Fig. 4 A-C).” Where are those Suppl. Figs?

All these figures are in the Supplementary file. This is volcano plot data.

  • Lines 358-359: “Prolonged administration of siRNA #5 lead to phenotype changes in vivo close to those in vitro.” What did they mean “to phenotype changes in vivo” in what ways? Any data?

By this sentence we generalized the results of transcriptome data analysis by PANTHER (Fig. 5).

  • Lines 404-405: “more active siRNA #5 caused more significant changes in gene expression than siRNA #7 both in vitro and in vivo (Fig. 4).” - They said multiple, but only two listed – what happened to others?

As demonstrated on Fig. 4 siRNA #5 caused the reliable changes in more than 10 PANTHER pathways in the liver. In comparison to this data siRNA#7 influenced only on two PANTHER pathways after 6 days of the administration and four – after 13 days with a statistically significant p-value (<0,05).

  • Lines 436-437: “Here we demonstrated protein level-dependent effects on DDX3 RNA helicase knockdown in the murine liver.” What was about the RNA seq data set?

We changed the sentence in the text.

  • Precaution is needed for spelling and English expression patterning, e.g., title “Eccessive suppression of murine DDX3 RNA helicase is negative for hepatocytes in vitro and in vivo” should be Excessive.

Thank you for the recommendation. We additionally checked English in our paper.

Reviewer 2 Report

In their manuscript, Sergeeva et al. examine the effects of two shRNAs against DDX3 in cultured cells (Hepa 1-6) and in vivo in mouse hepatocytes.  They characterize the knockdown efficiency, examine the effects on cell proliferation and apoptosis, and perform RNA-seq experiments to analyze changes in gene expression.  Overall, the experiments they perform are executed well, however, there are a few major limitations in their manuscript that would need to be addressed before publication.

Major issues:

1) The authors claim that knockdown of DDX3 in the murine liver induces apoptosis, yet the only data presented is H&E staining to examine the morphology of the cells.  While the authors can observe dead cells, to support the claim of apoptosis, the authors perform experiments to show that the cells are dying through apoptosis specifically, and not through another pathway (e.g. staining for markers of apoptosis).  The staining for markers of apoptosis and quantifying the staining can also help provide more data to support the claim that long-term treatment with siRNA #7 caused less cell death, as the evidence presented for this claim is limited and non-quantitative.

2) The RNA-seq data is lacking validation, especially claims about certain pathways being affected by DDX3 knockdown.  The authors should conduct additional experiments to validate the results from their RNA-seq analysis.  For example, the authors can RT-qPCR a few key genes to show that the changes in the expression of these genes are consistent with what they observed in the RNA-seq experiment, and they should look at some markers for some of the important signaling pathways they note (e.g. the cell cycle, Wnt, and cadherin pathways that were important enough to highlight in the abstract) to confirm that these pathways are deregulated as expected from the RNA-seq data.

3) The authors attribute the differences between siRNAs #5 and #7 to the different knockdown efficiencies observed.  However, an alternative explanation that the authors have not ruled out is that the siRNAs have different off target effects that, at least in part, contribute to the different phenotypes observed.  The authors should discuss this caveat in the discussion.

Minor issues:

1) For the MTS assay in Fig 3a, I would prefer to see the growth curves for the siControl, si5 and si7 samples plotted separately rather than showing the si5 and si7 data plotted relative to siControl.

2) The authors refer to the plots in Fig 4A as volcano plots, but these are not volcano plots. Volcano plots would typically plot the significance (e.g. -log(p-value)) of a differentially expressed gene against the log(fold-change) in expression observed in a differential expression experiment.  Furthermore, the authors say that the plots show huge differences between the phenotypes of the cells, but I see the plots as saying that the two siRNAs affect most genes in a very similar way (i.e. there is a strong positive correlation between the FC with both siRNAs). 

Author Response

Thank you for your evaluation and providing many valuable comments and suggestions. We addressed all your recommendations as explained below. All corrections indicated with yellow.

In their manuscript, Sergeeva et al. examine the effects of two shRNAs against DDX3 in cultured cells (Hepa 1-6) and in vivo in mouse hepatocytes.  They characterize the knockdown efficiency, examine the effects on cell proliferation and apoptosis, and perform RNA-seq experiments to analyze changes in gene expression.  Overall, the experiments they perform are executed well, however, there are a few major limitations in their manuscript that would need to be addressed before publication.

Major issues:

1) The authors claim that knockdown of DDX3 in the murine liver induces apoptosis, yet the only data presented is H&E staining to examine the morphology of the cells.  While the authors can observe dead cells, to support the claim of apoptosis, the authors perform experiments to show that the cells are dying through apoptosis specifically, and not through another pathway (e.g. staining for markers of apoptosis).  The staining for markers of apoptosis and quantifying the staining can also help provide more data to support the claim that long-term treatment with siRNA #7 caused less cell death, as the evidence presented for this claim is limited and non-quantitative.

 We performed additional experiments and provided the data on Fig. 2B.

2) The RNA-seq data is lacking validation, especially claims about certain pathways being affected by DDX3 knockdown.  The authors should conduct additional experiments to validate the results from their RNA-seq analysis.  For example, the authors can RT-qPCR a few key genes to show that the changes in the expression of these genes are consistent with what they observed in the RNA-seq experiment, and they should look at some markers for some of the important signaling pathways they note (e.g. the cell cycle, Wnt, and cadherin pathways that were important enough to highlight in the abstract) to confirm that these pathways are deregulated as expected from the RNA-seq data.

 We performed additional experiments and provided the data on the Fig. 6.

3) The authors attribute the differences between siRNAs #5 and #7 to the different knockdown efficiencies observed.  However, an alternative explanation that the authors have not ruled out is that the siRNAs have different off target effects that, at least in part, contribute to the different phenotypes observed.  The authors should discuss this caveat in the discussion.

We added the additional explanation in the “Discussion” section.

 Minor issues:

  • For the MTS assay in Fig 3a, I would prefer to see the growth curves for the siControl, si5 and si7 samples plotted separately rather than showing the si5 and si7 data plotted relative to siControl.

We changed the representation of MTS assay.

  • The authors refer to the plots in Fig 4A as volcano plots, but these are not volcano plots. Volcano plots would typically plot the significance (e.g. -log(p-value)) of a differentially expressed gene against the log(fold-change) in expression observed in a differential expression experiment.  Furthermore, the authors say that the plots show huge differences between the phenotypes of the cells, but I see the plots as saying that the two siRNAs affect most genes in a very similar way (i.e. there is a strong positive correlation between the FC with both siRNAs). 

We changed the figure description and a bit modified the text to explain our conclusions.

Round 2

Reviewer 1 Report

Specific comments:

  • Fig 2 needs a title before panel A & B legends. The authors need to follow the publication standard, i.e., putting panel letter-A and -B on the top left side of the designated panel, not the below the panel as shown in the current version. They need to rearrange all the panel letters among Fig 3, Fig 4, Fig 6, as well.

Author Response

Thank you for your evaluation of the manuscript and providing valuable comments and suggestions. We corrected the formatting of the figures and added the legend. All our corrections in the text indicated in light blue.

Reviewer 2 Report

Sergeeva et al. have addressed most of my concerns, and the manuscript is much improved.  I still have a few concerns that should be addressed before the manuscript is suitable for publication.

Major concerns:

  1. I’m somewhat unclear about the controls presented throughout the manuscript. For example, in Fig 1, the control is shown separate for the 6 day, 8 day and 13 day samples and are labeled with si5 and si7.  Are the controls treated with si5 and si7?  From the methods, it seems like the control are samples treated with an siRNA targeting luciferase.  If that is the case, the authors should show the results from the control sample at the same timepoints as they show for the siDDX3 samples (e.g. for Fig 1B, show results for siControl, si5 and si& for all three time points).  This also applies to Fig 1C, Fig 2B, and Fig 3B.
  2. I do not believe that the data presented in this manuscript can rule out the possibility that different off target effects of the two siRNAs contribute to some of the different phenotypes observed when comparing the two siRNAs. Although the authors added a discussion of off target effects in the discussion, they should explicitly acknowledge this limitation and state that they cannot rule out the possibility that some of the observed differences in phenotype could be attributed to different off target effects of the two siRNAs.

Minor concerns:

  1. In the methods, the authors write that sequencing was performed in 50 nt single read mode, but the analysis refers to paired-end reads in lines 190 and 191.

Author Response

Thank you for your evaluation of the manuscript and providing valuable comments and suggestions. We addressed all recommendations as explained below. All our corrections in the text indicated in light blue.

Major concerns:

  1. I’m somewhat unclear about the controls presented throughout the manuscript. For example, in Fig 1, the control is shown separate for the 6 day, 8 day and 13 day samples and are labeled with si5 and si7.  Are the controls treated with si5 and si7?  From the methods, it seems like the control are samples treated with an siRNA targeting luciferase.  If that is the case, the authors should show the results from the control sample at the same timepoints as they show for the siDDX3 samples (e.g. for Fig 1B, show results for siControl, si5 and si& for all three time points).  This also applies to Fig 1C, Fig 2B, and Fig 3B.

  • In all experimental work in vitro and in vivo we used siRNA targeting luciferase as a non-targeting control. We performed preliminary experiments and found that mRNA and protein levels of DDX3 are stable under siLuc treatment in vitro and in vivo. We added this data in the Suppl. Fig. 1G-I and corrected the text.

  1. I do not believe that the data presented in this manuscript can rule out the possibility that different off target effects of the two siRNAs contribute to some of the different phenotypes observed when comparing the two siRNAs. Although the authors added a discussion of off target effects in the discussion, they should explicitly acknowledge this limitation and state that they cannot rule out the possibility that some of the observed differences in phenotype could be attributed to different off target effects of the two siRNAs.

  • Thank you. We additionally discussed this point. This issue is fundamental for any transient knockdown system, so conclusive evidence is impossible. However, we used several approaches to confirm that we observe on-target effect. At the same time, we agree that we can’t exclude that some effects are still due to off-targets.

Minor concerns:

  1. In the methods, the authors write that sequencing was performed in 50 nt single read mode, but the analysis refers to paired-end reads in lines 190 and 191.
  • Thank you. We corrected this in the text.